# Genome-Wide Identification and Characterization of Homeobox Transcription Factors in *Phoma sorghina* var. *saccharum* Causing Sugarcane Twisted Leaf Disease

**DOI:** 10.3390/ijms25105346

**Published:** 2024-05-14

**Authors:** Yixue Bao, Jinlan Deng, Sehrish Akbar, Zhenzhen Duan, Chi Zhang, Wenfeng Lin, Suyan Wu, Yabing Yue, Wei Yao, Jianlong Xu, Muqing Zhang

**Affiliations:** 1State Key Lab for Conservation and Utilization of Subtropical Agric-Biological Resources & Guangxi Key Lab for Sugarcane Biology, Guangxi University, Nanning 530004, China; baoyixue57319@163.com (Y.B.);; 2Institute of Crop Sciences, Chinese Academy of Agricultural Sciences, Beijing 100081, China

**Keywords:** *Phoma sorghina* var. *saccharum*, genome, homeobox transcription factor, gene knockout, pathogenicity

## Abstract

A homeobox transcription factor is a conserved transcription factor, ubiquitous in eukaryotes, that regulates the tissue formation of structure, cell differentiation, proliferation, and cancer. This study identified the homeobox transcription factor family and its distribution in *Phoma sorghina* var. *saccharum* at the whole genome level. It elucidated the gene structures and evolutionary characteristics of this family. Additionally, knockout experiments were carried out and the preliminary function of these transcription factors was studied. Through bioinformatics approaches, nine homeobox transcription factors (*PsHOX1–PsHOX9*) were identified in *P. sorghina* var. *saccharum*, and these contained HOX-conserved domains and helix–turn–helix secondary structures. Nine homeobox gene deletion mutants were obtained using the homologous recombinant gene knockout technique. Protoplast transformation was mediated by polyethylene glycol (PEG) and the transformants were identified using PCR. The knockouts of *PsHOX1*, *PsHOX2*, *PsHOX3*, *PsHOX4*, *PsHOX6*, *PsHOX8*, and *PsHOX9* genes resulted in a smaller growth diameter in *P. sorghina* var. *saccharum*. In contrast, the knockouts of the *PsHOX3*, *PsHOX6*, and *PsHOX9* genes inhibited the formation of conidia and led to a significant decrease in the pathogenicity. This study’s results will provide insights for understanding the growth and development of *P. sorghina* var. *saccharum*. The pathogenic mechanism of the affected sugarcane will provide an essential theoretical basis for preventing and controlling sugarcane twisted leaf disease.

## 1. Introduction

Sugarcane (*Saccharum* spp.) is an important crop that grows in tropical and subtropical regions [1]. Sugarcane production is affected by many abiotic and biotic factors. Biotic stresses include viral, fungal, and bacterial diseases. The genus *Phoma* is one of the most abundant fungal genera in the world and its members are ubiquitous in the environment [2]. It can parasitize plants, animals, and even human skin [3,4]. Most of the species are found associated with land plants, causing mainly leaf and stem blotches [2]. Sugarcane twisted leaf disease is caused by a fungal pathogen, *Phoma* sp., which shows curling and twisting symptoms of sugarcane crown leaves with yellowing midribs [5]. Curly leaves break the leaf flatness and influence physiology in sugarcane. This disease was first reported in 2014 in Guangxi, China, and this resulted in a 5~10% yield loss and became a potential threat for the sugarcane industry [5]. Subsequently, the researcher continued to conduct in-depth research and finally named the causal agent of the sugarcane leaf curl disease in China as *Phoma sorghina* var. *saccharum* (*P. sorghina* var. *saccharum* as the abbreviation) [6].

High-throughput sequencing has obtained significant attention as the most reliable and effective technology to analyze genetic mechanisms [7]. Due to the advancement of sequencing technology and bioinformatics, exploring fungal development, systematics, and evolution has become more feasible at the molecular level [8]. We briefly reported the genome of *P. sorghina* var. *saccharum* BS2-1 [9]. Three draft genome sequences of the *Phoma* sp. are also available, including *Phoma herbarum* JCM 15942 (unpublished, BCGR00000000), *Phoma* sp. RAV-16-625 (unpublished, SAUA00000000), and *Phoma* sp. XZ068 [10].

The homeobox transcription factor (HOX) is a class of transcriptional regulators with conserved structure and function [11]. The homeobox protein can bind to the regulatory sequence in target genes to affect their transcription [12], which regulates the tissue formation of structure, cell differentiation and proliferation, and eukaryote carcinogenesis [13]. The HOX gene family has highly conserved domains, called homeodomains (HDs), consisting of 60 or 61 amino acids [14,15]. *HOX* genes was first identified in fungi from the *Setospheria turcica* genome through transcriptome analysis [16]. After that, it was identified in many other fungal species including the pathogens causing rice blast (*Magnaporthe oryzae*) [17], cucumber anthracnose (*Colletotrichum orbiculare*) [18], and anthracnose disease on pepper fruit (*Collatotrichum scovelli*) [19]. The HOX family gene of fungi is mainly involved in regulating mycelial polar growth, sexual reproduction, sporulation, and pathogenicity [20,21]. According to research findings, deleting the homeobox gene *GRF10* in *Candida albicans* inhibits mycelial growth, spore defects, and biofilm formation [22]. Moreover, the HD-Zip family might regulate the leaf curled and twisted [23,24,25].

The objectives of the current study were as follows: the (i) in-depth molecular and genomic analysis of twisted-leaf-disease pathogen *P. sorghina* var. *saccharum*, (ii) identification and characterization of the PsHOX gene family, and (iii) functional analysis of the PsHOX gene family through mutation. For this purpose, the assembly and annotation of the *P. sorghina* var. *saccharum* BS2-1 complete genome revealed the presence of non-coding RNAs and repetitive elements, a biosynthetic gene cluster, and transcription factor analysis involving *Phoma* species.

## 2. Results

### 2.1. Assessment of Genome Assembly and Completeness

To obtain a high-quality BS2-1 genome of *P. sorghina* var. *saccharum*, we incorporated multiple sequencing technologies, including Illumina and PacBio. These technologies generated 3.2 Gb of long reads with N50 of 21,441 bp, covering 97× of the estimated genome size. A total of 33.12 Mb genomic sequences were obtained with the contig N50 of 1.92 Mb and GC content of 52.12% (Appendix A). The genome sequences obtained were submitted to the GenBank database (accession number VXJJ00000000). The assembly process generated 22 contigs. The shortest was 28,739 bp observed for contig29. However, the longest was 4,023,921 bp in contig00. Moreover, the size of the mitochondrial was about 66,703 bp. All the contigs possessed 3′ telomere repeats except for contig 29. A 5′ telomere repeat was found in contig00, 01, 08, 10, 13, 16, 17, 19, 21, 23, and 25 (Appendix A).

BUSCO was used to assess the quality of genome assembly. BUSCO analysis for *P. sorghina* var. *saccharum* BS2-1 identified two-hundred-and-eighty-six (98.6%) complete and single-copy, one (0.3%) fragmented, and three (1.1%) missing genes from a total of two-hundred-and-ninety BUSCO groups (Appendix A), suggesting that the BS2-1 genome assembly is highly complete. Additionally, we conducted a comparison of these assemblies with *Phoma* sp. XZ068, *Phoma herbarum* JCM 15942, and *Phoma* sp. RAV-16-625. The contig N50 of the BS2-1 assemblies reached 1.92 Mb, surpassing the sizes of XZ068 (1.50 Mb), JCM 15942 (0.56 Mb), and RAV-16-625 (0.20 Mb). The contig count was 22, significantly lower than the previous assembly counts of 33 (XZ068), 498 (JCM 15942), and 383 (RAV-16-625). Overall, the assembled genome represents the highest quality achieved to date.

### 2.2. Gene Annotation

A combination of *ab initio* gene prediction and homologous protein evidence led to the annotation of 7973 protein-coding genes in the BS2-1 genome. The average gene length was 2219 bp, with an average exon length of 595.3 bp and an average intron length of 212.6 bp (Appendix A). Of these genes, 7854 (98.5%) genes were annotated in the TrEMBL database, 7851 (98.5%) in the NR database, 6169 (77.4%) in the Pfam database, 5239 (65.7%) in the Swiss-Prot database, 4415 (55.4%) in the GO database, 3556 (44.6%) in the COG database, and 2767 (34.7%) in the KEGG database. Thereinto, 2440 (30.6%) genes were annotated in the PHI database, 515 (6.5%) in the CAZyme database, and 115 (1.4%) in the TCDB database. We also identified 583 (7.3%) secreted proteins, 168 (2.1%) tRNAs, 113 (1.4%) rRNAs, and 76 (1.0%) pseudogenes (Appendix A). A total of 411 repeat sequences were identified, occupying a length of 207,578 bp and genome coverage of 0.63%. The maximum number of repeat sequences identified constituted LTR/Gypsy (158), followed by TIR/Tc1-Mariner (57) and SINE (50) (Appendix A). In total, 147 transcription factors (TFs) were identified in the BS2-1 genome, including 39 zf-C_2_H_2_ and 19 MYBs (Appendix A). The quality of this genome is more significant than previously published genomes. We located the telomere regions in the chromosomes and the structure and function of the genome were analyzed in detail, including but not limited to functional details annotated in the public and specific databases and identified in repeat sequences.

### 2.3. Identification of PsHOX Gene Family

Nine homeobox genes were identified from the whole genome of BS2-1, and these were named *PsHOX1*–*PsHOX9*. The family members were distributed in six contigs using the genome sequence and annotation information (Appendix A). Among them, *PsHOX1* and *PsHOX2* were localized on contig04; *PsHOX3* on contig19; *PsHOX4* and *PsHOX9* on contig00; *PsHOX5*, *PsHOX6*, and *PsHOX8* on contig02; and *PsHOX7* on contig03. The predicted physicochemical properties of HOX proteins indicated that these nine proteins were different in terms of amino acids, molecular weight, and theoretical PIs. The length of the protein encoded by each gene ranged from 448 to 1160 aa, and the molecular weight ranged from 57,846.51 to 133,759.79. Most of these proteins are acidic while only three genes encoded proteins (*PsHOX2*, *PsHOX4*, and *PsHOX9*) that were alkaline. Furthermore, these proteins are unstable (instability index > 40). Sub-cellular localization prediction found that eight were localized in the nucleus, and only *PsHOX4* was localized in the nucleus or mitochondria (Table 1).

### 2.4. Characterization of PsHOX Gene Family and Evolutionary Analysis

Amino acid sequences of *PsHOX genes* (*PsHOX1–PsHOX9*) were aligned with *Magnaporthe oryzae MGG_04853*, *Saccharomyces cerevisiae SCRG_00601*, *Fusarium graminearum FGSG_07097*, *Neurospora crassa NCU09556*, *Aspergillus nidulans AN2009*, *Aspergillus terreus ATEG_00728*, and *Aspergillus flavus AFLA_03039*. The amino acid sequences of the homeodomains in these proteins were similar and had identical amino acid residues at the L, P, W, N, and R positions. The secondary structures of the proteins showed that these HOX proteins had a typical helical structure (helix–turn–helix). The five conserved amino acid residues were helix I, loop, and helix III. Comparative analysis showed that helix III had higher conservation (Figure 1A).

The structural analysis of the PsHOX gene family showed that all genes had introns, among which *PsHOX1*, *PsHOX5*, and *PsHOX6* contained one intron; there were two in *PsHOX2*, *PsHOX3*, and *PsHOX4*; there were three in *PsHOX7*; and there were four in *PsHOX8* and *PsHOX9*. All nine genes contained a homeodomain, but their positions were different. *PsHOX1*, *PsHOX2*, *PsHOX3*, *PsHOX5*, *PsHOX6*, *PsHOX8*, and *PsHOX9* were near the N-terminus; *PsHOX7* was in the middle, and *PsHOX4* was near the C-terminus. In addition to the homeodomain, some genes had other domains, such as 1–3 tandem ZnF-C_2_H_2_ in *PsHOX5*, *PsHOX6*, *PsHOX7*, *PsHOX8*, and *PsHOX9*. The *cis*-acting regulatory element was also detected in the PsHOX family genes, including the auxin-responsive, light-responsive, meristem expression, salicylic acid-responsive, MeJA-responsive, gibberellin-responsive, circadian control, and abscisic acid-responsive (Figure 1B) genes. An evolutionary analysis of the HOX gene family was performed to compare *P. sorghina* var. *saccharum*, *N. crassa*, *A. nidulans*, *A. terreus*, *A. flavus*, *A. fumigatus*, *M. oryzae*, *S. cerevisiae*, and *F. graminearum*. The PsHOX gene family (*PsHOX1*–*PsHOX9*) was clustered into four clades. *PsHOX1*, *PsHOX6*, *PsHOX5*, *PsHOX8*, and *PsHOX9* were grouped in clade E, *PsHOX7* and *PsHOX2* in clade A and clade B, and *PsHOX3* and *PsHOX4* in clade C (Figure 1C).

### 2.5. Verification of Knockout Mutants

In order to study the function of genes encoding homeobox transcription factors in *P. sorghina* var. *saccharum*, we used a simple and efficient method (protoplast preparation, transformation, and gene knockout system) established by the research group to quickly obtain gene knockout mutants of pathogenic fungi [26]. The principle of linear DNA fragment fusion and homologous recombination knockout was used and hygromycin hyg replaced the *PsHOX1*–*PsHOX9* target gene. The knockout principle is shown in Appendix A, and the method is detailed in the Section 4. We picked the transformants, and their DNA was extracted; these extractions were further screened using PCR. When normal knockout mutants are amplified using PCR using *gene-OF*/*gene-OR* primers of the target gene, the corresponding ORF fragments were not amplified. However, when the *gene-AF/P816-R* primers were used for PCR amplification, the corresponding Hyg-AP fragment was obtained. The mutant could also obtain the corresponding Hyg-BP fragment with the *P625-F/gene-BR* primers. Using wild-type BS2-1 as a control, the corresponding ORF fragment was amplified using the *gene-OF/gene-OR* primers of the target gene. However, the corresponding fragment could not be amplified using *gene-AF/P816-R* or *P625-F/gene-BR* primers (Appendix A). PCR verification confirmed that nine *PsHOX* genes had successfully been knocked out in BS2-1.

### 2.6. PsHOX Gene Knockout: Effect on Growth and Conidia Formation

The growth of the mutants of *P. sorghina* var. *saccharum* in the PDA medium was affected after the *PsHOX* gene was knocked out. The wild-type-strain colony appeared to have a white villous, spherical, radial, and dense mycelium from the top while it was crimson at the bottom. The colony growth diameter had significantly reduced in the mutant strains compared to wild-type ones. The growth of the other seven mutants was affected in the colony diameter, except that of Δ*PsHOX5* and Δ*PsHOX7*. Noticeably, the maximum reduction in colony size was detected in Δ*PsHOX3*, Δ*PsHOX6*, and Δ*PsHOX9*. The deletion of *PsHOX6* severely affected the average growth of the strain. In addition, the knocking out of the *PsHOX6* and *PsHOX9* genes resulted in a marked decrease in pigment production, and the color of the produced pigment changed from dark red to light pink in the outer ring (Figure 2). This study further observed the conidia formation in the mutant. As per microscopy, the spore count was also meager in Δ*PsHOX3*, Δ*PsHOX4*, Δ*PsHOX6,* and Δ*PsHOX9*, indicating that the knocking out of the *PsHOX* gene may led to changes in the sporulation structure of *P. sorghina* var. *saccharum*, thereby inhibiting the formation of spores. The deletion of *PsHOX* genes leads to a decrease in the sporulation of *P. sorghina* var. *saccharum* strains.

### 2.7. Knockout of PsHOX Genes for the Pathogenicity of P. sorghina var. saccharum

The pathogenicity experiment showed significant differences in the pathogenicity between the wild type and mutants. The disease symptoms of sugarcane leaves inoculated with BS2-1 showed that the leaves were obvious twisted and curled. However, the sugarcane leaves inoculated with nine mutants showed mild leaf disease symptoms and the loss of twisted symptoms (Figure 3). We also observed the incidence of sugarcane inoculated with the wild-type and mutants. Real-time PCR of infected leaf samples depicted threshold values under 35, which are confidently considered infected (Appendix A). We have identified around 90% of the maximum incidence rate of wild-type BS2-1. However, the disease incidence of mutant strains was significantly reduced, with the lowest incidence rates being 20% (Δ*PsHOX3*), 10% (Δ*PsHOX6*) and 10% (Δ*PsHOX9*) (Appendix A), indicating that the deletion of the *HOX* gene had significantly affected the pathogenicity of the mutant strains. In addition, the gDNA of the nine complementary strains was used as a template for PCR verification. The PCR results and phenotype showed that complementary strains were consistent with the wild-type strain BS2-1 (Appendix A).

### 2.8. PsHOX Gene Knockout: Effect on Secondary Metabolisms of P. sorghina var. saccharum

In order to deeply understand the impact of *PsHOX* genes on the synthesis of secondary metabolites, LC-MS/MS non-targeted metabolomics was performed on the mycelia of wild-type and four mutant strains (Δ*PsHOX1*, Δ*PsHOX3*, Δ*PsHOX6*, Δ*PsHOX9*). The total ion chromatogram was measured (Appendix A). The results show that the TIC overlap in the positive and negative ion modes was high, the detected substance peaks were relatively abundant, and the changing trends of the total ion chromatograms under different conditions were different. For example, the retention time was between 11 and 16 min, the peaks appearing under different conditions corresponded to the types, and the relative contents of substances were obviously different. After LC-MS/MS raw data preprocessing, multiple databases were used for search and matching (see Section 4). A total of 161 secondary metabolites were detected, divided into 13 categories, with Fatty Acyls being the most common (sixteen species), followed by carboxylic acids and derivatives (fifteen species), Organooxygen compounds (thirteen species), Benzene and substituted derivatives (ten species), and Pyridines and derivatives (seven species) (Table 2).

Based on Student’s *t*-test *p* < 0.05 and Fold change > 2, the screening of differentially expressed metabolites (DEMs) was performed on the four groups of test samples. The results showed that the total numbers of significantly DEMs in Δ*PsHOX1* vs. WT, Δ*PsHOX3* vs. WT, Δ*PsHOX6* vs. WT, and Δ*PsHOX9* vs. WT were 6, 30, 32, and 36, respectively. Among them, the Δ*PsHOX1* group had very few significantly DEMs while the Δ*PsHOX3*, Δ*PsHOX6*, and Δ*PsHOX9* groups had more, mainly including Choline, D-Ribose-1-phosphate, NAD+, α,α-Trehalose, β-Nicotinamide mononucleotide, etc. Among them, the metabolite with the largest difference was α,α-Trehalose, with a difference of 157 times. These DEMs indicate that the knocking out of *PsHOX3*, *PsHOX6*, and *PsHOX9* genes inhibits the production of *P. sorghina* var. *saccharum* metabolites, thereby indirectly affecting the growth and development, conidia formation, and pathogenicity of the strain.

## 3. Discussion

Sugarcane twisted leaf disease is caused by *Phoma sorghina* var. *saccharum*. Our research presented here described the detailed features of the BS2-1 genome in detail, then identified and characterized the disease-associated transcription factors. The BS2-1 genome was assembled onto 22 contigs, with a contig N50 of 1.92 Mb, and the final genome size was 33.12 Mb. The quality of this genome was better than that of previously published genomes, such as *Phoma* sp. XZ068; a total of 33 contigs were generated with N50 of 1.50 Mb [10]. A 98.6% of complete BUSCOs suggested that our assembly of the BS2-1 genome was relatively accurate and complete.

The homeobox protein encoded by the homeobox gene, as a type of transcriptional regulatory factor, plays an important role in the regulation of ontogeny and cell differentiation in eukaryotes. So far, homeobox gene families have been identified in animals [27], plants [28] and fungi [20,29], but the types and numbers of their family members are different. There are 16 categories of homeobox genes found in animals and 11 categories in plants [30]. Nine *PsHOX* genes were identified in the model organism *Saccharomyces cerevisiae* [31], twelve were identified in *Fusarium graminearum* [32], and eight were identified in *Magnaporthe oryzae* [17]. This study identified nine *PsHOX* genes in the BS2-1 genome. There are large differences in the types and numbers of HOX gene families among the above-mentioned different biological groups and species. This phenomenon could be attributed to the varying evolutionary positions of the species themselves. Generally, higher species, such as animals and plants, tend to have larger gene family sizes, reflecting greater complexity. Conversely, lower species, like yeast and filamentous fungi, typically exhibit smaller gene family sizes. This observation underscores the crucial role that this family plays in regulating the growth and development across diverse species.

The research also found that the members of the homeobox gene family found in *S. cerevisiae* and filamentous fungi contain a varying number of conserved homeodomain and ZnF-C_2_H_2_ domains. *Ascomycetes* subclass *Hypocreomycetidae* contains more homeobox-C_2_H_2_ genes, with species such as *Verticillium dahliae* and *Fusarium oxysporum* both identified as having seven homeobox-C_2_H_2_ genes [33]. However, *Sordariomycetidae* and *Leotiomycetes* contain fewer homeobox-C_2_H_2_ genes. For example, *M. oryzae*, *N. crassa*, and *S. sclerotiorum* contain only one homeobox-C_2_H_2_ gene while *S. cerevisiae* does not have homeobox-C_2_H_2_ genes [17,34]. Our results showed that five out of nine homeobox genes had the ZnF-C_2_H_2_ zinc finger domain in *P. sorghina* var. *saccharum*, including *PsHOX5*, *PsHOX6*, *PsHOX7*, *PsHOX8*, and *PsHOX9.* The homeobox-C_2_H_2_ gene (*MGG_01730*) containing the ZnF-C_2_H_2_ domain in *M. oryzae* does not affect the growth and pathogenicity of bacteria [17]. However, *PsHOX6* and *PsHOX9,* with similar structures in *P. sorghina* var. *saccharum,* were involved in mycelial growth and played an essential role in forming conidia in our study. It can be seen that the difference in the number of homeobox-C_2_H_2_ genes contained in different fungi is closely related to the evolutionary level of the species. The reason for this phenomenon may be due to gene duplication events. From the perspective of functional analysis, the conserved Homeobox-C_2_H_2_ structure may regulate different metabolic processes by regulating different target genes. However, the specific mechanism requires an in-depth study.

In recent years, multiple homeobox genes have been reported in fungi; the homeobox gene in fungi is mainly involved in regulating polar growth, sporulation, and pathogenicity [20,21]. In order to study the function of homeobox genes in *P. sorghina* var. *saccharum*, this study conducted an in-depth analysis on nine *PsHOX* genes’ functions and found that *PsHOX* genes are involved in the physiological processes such as development, asexual reproduction, and the synthesis of secondary metabolites.

*HOX* transcription factors play an essential role in fungal mycelial growth and development. The specific expression of the homeobox transcription factor *WUS* in the shoot apical meristem can promote the proliferation of tissue cells in specific areas [35]. In *C. albicans*, the HOX gene *GRF10* regulates mycelial growth [22]. The knocking out of the HOX gene *pah1*, which was first reported in *Podospora anserina*, resulted in reduced hyphal growth and abnormal colony morphology. Microscopic observation revealed that the aerial hyphal tips of the mutant appeared to be excessively branched [36]. The above results indicate that the role of homeobox transcription factors in regulating fungal hyphal development is relatively conserved. This study found that the knocking out of the *PsHOX1*, *PsHOX2*, *PsHOX3*, *PsHOX4*, *PsHOX6*, *PsHOX8*, and *PsHOX9* genes resulted in a smaller growth diameter of the mutants on PDA. The growth defects of mutants lacking the *PsHOX3*, *PsHOX6* and *PsHOX9* genes were most obvious. This shows that the PsHOX gene is involved in the vegetative growth of *P. sorghina* var. saccharum and plays an important role in the growth and development of hyphae.

In addition to the important regulation of hyphal development, *HOX* transcription factors also play an important role in the asexual reproduction of fungi. After the deletion of *HOX2* in *F. graminearum*, the development and synthesis of conidia were inhibited [32]. *HOX2* also plays a crucial role in the spore development of *M. oryzae*, and the loss of *MoHOX2* causes *M. oryzae* to lose the ability to form conidia [17,20]. Similar results have been reported in *C. scovillei*, where four out of ten *HOX* genes (*CsHOX1*, *CsHOX3*, *CsHOX4*, and *CsHOX5*) are associated with morphology and conidial growth size [19]. In this study, we found that the deletion of the *PsHOX3*, *PsHOX4*, *PsHOX6*, and *PsHOX9* genes inhibited the formation of conidia, and the deletion of *PsHOX3*, *PsHOX6*, and *PsHOX9* resulted in a significant reduction in the number of conidia produced by the mutants, further indicating that these three *PsHOX* genes (*PsHOX3*, *PsHOX6*, and *PsHOX9*) in *P. sorghina* var. *saccharum* are essential in regulating conidia formation.

Although *HOX* transcription factors are increasingly studied in fungi, there is not much information on the regulation of secondary metabolites by *HOX*. This study found that *PsHOX3*, *PsHOX6*, and *PsHOX9* play equally important roles in the production of *P. sorghina* var. *saccharum* metabolites. Knocking out these three genes also resulted in inhibited metabolite production in the mutant strain, particularly α,α-Trehalose. Previous studies have shown that α,α-Trehalose is widely present in various organisms such as bacteria, yeast, filamentous fungi, plants, and invertebrates. It serves as a reserve of energy and carbon sources, a signal-sensing complex, and a growth regulatory factor, playing a crucial role in organismal growth and development [37]. The knocking out of *HOX* transcription factors significantly inhibits the production of α,α-Trehalose, directly impacting the growth and development of the strain.

The knocking out of homeobox transcription factors also affected the pathogenicity and disease phenotype of *P. sorghina* var. *saccharum*. The knocking out of *PsHOX1–PsHOX9* led to the different levels of reduction in the pathogenicity of the mutants, and the incidences of the sugarcane inoculated by Δ*PsHOX3*, Δ*PsHOX6*, and Δ*PsHOX9* were only 20%, 10%, and 10%. Notably, sugarcane plants inoculated with wild-type BS2-1 showed twisted and curled leaves. However, sugarcane plants inoculated with deletion mutants did not show typical symptoms of sugarcane twisted leaf disease. Leaf shape is determined by the behavior and state of leaf cells, so some gene mutations that affect the state of cell division, growth, and differentiation will cause leaf curl to a large extent. Previous studies showed that *PHV*, *PHB*, and *REV* in the HD-Zip family are genes regulating leaf adaxial planarization [23,24]. The miR165/166 target gene is a member of the HD-Zip family. *ICU4* is also a member of the HD-Zip family, and its double mutants with *ago1*, *hen1*, and *hst* have more severe leaf curl [25]. Therefore, it is speculated that the homeobox transcription factor might play a significant regulatory role in curling and twisting sugarcane leaves (Figure 4). The knocking out of the homeobox transcription factor reduced sugarcane’s typically twisted leaf disease.

## 4. Materials and Methods

### 4.1. Isolation of Pathogen Causing Sugarcane Twisted Leaf Disease

Sugarcane leaf samples showing twisted leaf disease symptoms were collected from the field of Guangxi, China (Appendix A). The wild-type strain BS2-1 was isolated from leaf samples and cultured on the Potato Dextrose Agar (PDA) medium (Hopebio, Qingdao, China) at 28 °C as in previous report [5].

### 4.2. Genome Sequencing, Assembly, and Evaluation

The gDNA was extracted using the CTAB method [38]. DNA quality and concentration were measured using a Qubit Fluorometer spectrophotometer (Thermo Fisher Scientific, Waltham, MA, USA). The third-generation sequencing of BS2-1 was performed based on the sequencing platform Illumina (Illumina, San Diego, California, USA) and PacBio (PacBio, Menlo Park, California, USA). The filtered subread data were assembled using the *Canu* v.2.0 software to obtain the complete genome [39]. Genome was evaluated for completeness and accuracy based on BUSCO (Benchmarking Universal Single-Copy Orthologs) [40], which is the eukaryotic database (eukaryota_odb9) that contains 290 conserved core genes in fungi. The repeat sequence database of the BS2-1 genome was constructed based on LTR_FINDER [41], MITE-Hunter [42], RepeatScout [43], and PILER-DF [44]. The database was classified by PASTEClassifier v.2.0 [45] and then combined with Repbase [46]. The BS2-1 was deposited in the final repeat sequence database. RepeatMasker v.2.0 software was used to predict the repeat sequence of BS2-1 based on the constructed repeat sequence database [47].

### 4.3. Gene Prediction and Genome Annotation

The gene structure prediction of BS2-1 was performed using Augustus v.3.2.3, GlimmerHMM v.3.0.4, and SNAP for *de novo* prediction [48,49,50]. GeMoMa v.1.6 was used for the prediction based on homologous species [51]. EVM was used to integrate the above two prediction results [52]. The rRNA was predicted using Infernal v.1.1 [53] while tRNAscan-SE v.2.0 was used to identify tRNA [54]. The predicted gene sequences were blasted against the available databases, such as COG [55], KEGG [56], Swiss-Prot, TrEMBL [57], and Nr [58,59], to obtain gene function annotation results. Based on the Nr database alignment results, the Blast2GO [60] and hmmer [61] were used to perform functional annotation of the GO [62] and Pfam [63] databases (access date: 10 February 2022).

### 4.4. Functional Gene Analysis

The protein sequences of the predicted genes were blasted and annotated against the TCDB and PHI databases [64,65]. Functional annotation of carbohydrate genes was performed using the hmmer, based on the CAZyme database [66]. SignalP 4.0 and tmhmm were used for signal peptide and transmembrane region prediction [67,68]. The protein containing the transmembrane helix was removed from the protein containing the signal peptides, and the remaining protein was the secreted protein (SP). The iTAK program was used to identify transcription factors (TFs), protein kinases (PKs), and transcription regulators (TRs) among the gene models [69] (access date: 20 February 2022).

### 4.5. Identification of the PsHOX Family Gene

The Hidden Markov Model of the HOX family gene was obtained via an online search against the Pfam database [63]. The hmmsearch searched the protein sequence of the whole genome in BS2-1. The *e*-value was set to default arguments. The SMART (http://smart.embl-heidelberg.de/, accessed on 10 December 2022) program was used to analyze the domains. The conserved motif sequences, including the HOX domain, were selected as candidate members. ExPASy-ProtParam (http://web.expasy.org/protparam/, accessed on 10 December 2022) was used to analyze the physicochemical properties of the HOX protein obtained in BS2-1 including the number of amino acids, molecular weight, theoretical PI, and instability index. Sub-cellular localization of all proteins was predicted via the online software WoLF-PSORT (http://www.genscript.com/psort/wolf_psort.html, accessed on 10 December 2022).

### 4.6. Structural Characteristics and Phylogenetic Analysis

The CDS (coding sequence) and gDNA sequence of the HOX family gene were obtained by searching the BS2-1 genome. The gene structure map was drawn using the GSDS 2.0 (http://gsds.gao-lab.org/index.php, accessed on 11 December 2022). *Cis*-acting regulatory element prediction was performed using PlantCare (http://bioinformatics.psb.ugent.be/webtools/plantcare/html/, accessed on 11 December 2022) and visualized using TBtools [70]. The amino acid sequences of the individual HOX domains were then aligned using Clustal X v.2.1 [71]. Phylogenetic trees were constructed using the neighbor-joining method in the MEGA v.7.0, with a bootstrap value of 1000 [72].

### 4.7. Construction of Homologous Replacement Fragments

The wild-type BS2-1 samples were used as templates. The gene replacement strategy was performed to displace the *PsHOX* gene with the *hygromycin* gene (Appendix A). The upstream left-border and the downstream right-border sequences of the *PsHOX1-9* were amplified using specific primers containing the splicer of the hygromycin gene fragment (Appendix A). The amplified gene was then eluted from the gel using a gel extraction kit (DP214-03, QIAGEN, Dusseldorf, Germany). The vector containing the hygromycin gene fragment was transformed into *E. coli*. The plasmid was extracted using a plasmid extraction kit (DP103-03, QIAGEN, Dusseldorf, Germany), and the 2145 bp hygromycin gene fragment was amplified using primers (*Hyg-F*/*Hyg-R*). According to the molar mass ratio of 1:3:1, the fragments of the left border, the hygromycin, and the right border were fused in proportion to adjust the total concentration of the fused DNA. In the case of fusion PCR, the amplification of 25 µL was performed with the following components: 1/4 primers (10 mM), 1 µL template DNA, 2× Rapid *Taq* Master Mix (Vazyme, Nanjing, China), and ddH_2_O. The PCR product was subjected to 1% agarose gel electrophoresis (Thermo Fisher Scientific, Waltham, MA, USA), and the target fragment was eluted. The recovered product was concentrated to 10 µL, with a final concentration over 2 µg.

### 4.8. Preparation of Protoplasts

A mixed enzyme solution of driselase (c8100, Solarbio, Beijing, China) and lywallzyme (L8141, Solarbio, Beijing, China) was prepared. Spore solution (wild-type BS2-1) with a concentration of 1 × 10^7^ CFU/mL was added into PDW medium (100 mL) and incubated at 28 °C for 12~15 h (at 160 rpm). Separated mycelia were washed with KCl solution (1.2 M) (Solarbio, Beijing, China) several times. The mycelia were then placed into the prepared enzymolysis solution and incubated at 28 °C for 2.5~3.5 h (at 60 rpm). Mycelia were rinsed with KCl solution and centrifuged at 4 °C for 10 min at 4000 rpm. The washing process was repeated twice. The precipitated protoplasts were resuspended in 500 µL STC buffer (Solarbio, Beijing, China), added with 250 µL of PTC solution (Solarbio, Beijing, China), and counted with a hemocytometer.

### 4.9. Transformation

A total quantity of 125 µL of protoplasts was added to the exogenous fragment (2 µg DNA) and concentrated to 10 µL, then chilled on ice for 30 min. Then, PTC was mixed (1 mL) and placed in an incubator (28 °C) for 25 min. The suspension was centrifuged at 8000 rpm for 5 min (at 4 °C). The supernatant was discarded, and 500 µL of STC was added to resuspend the pellet (on ice). The suspension was again centrifuged at 8000 rpm (4 °C) for 1 min. The supernatant was discarded again, and 510 µL of STC was added and mixed via gentle pipetting. The protoplasts were extracted with a pipette and evenly plated on a sterile enzyme-free 90 mm plate. They were then quickly poured into 15 mL of TB_3_ regeneration medium, incubated at 48 °C for 4 h, and cultured at 28 °C. After 16 h of incubation, another 15 mL of TB_3_ regeneration medium with a 100 µg/mL concentration of hygromycin B was added and further incubated at 48 °C for 4 h. It was then cultivated in a 28 °C incubator until resistant transformants grew.

### 4.10. Screening and Identification of Resistant Transformants

Resistant plate screening was performed on the transformants, which grew on the third day. The grown transformants were transferred to PDA containing 50 µg/mL hygromycin (Solarbio, Beijing, China). The wild-type strain BS2-1 was used as a negative control. Positive transformants were grown on the hygromycin B plate. Resistant transformants were confirmed using PCR and purified through single-spore isolation.

### 4.11. Complementary Strain Construction

A gDNA fragment containing 1.5 kb upstream and 1 kb downstream of the target gene was amplified using PCR. The fragment was cloned into the pcpxG418 (fungal expression vector) containing a neomycin resistance marker. Gene fragments were fused with the remaining linear fragments (pgpd and tgpd) of the vector using homologous recombination. The recombinant plasmids were recovered according to the sizes of the fragments and then transformed into *E. coli* (DH5α) competent cells. Positive strains were verified using PCR to obtain functional complement mutants.

### 4.12. Phenotypic Observation and Pathogenicity Assay

The wild type and mutants were grown on PDA medium at 28 °C for 3~5 d under 12 h light/dark. For each sample, mycelium was collected, and conidia suspension (1 × 10^7^ CFU/mL) was prepared. Morphological characteristics of the colonies were observed on the medium. The number of sporulations was counted with a hemocytometer. The sugarcane variety Zhongzhe1 was selected and planted in 55 pots, each containing one stem segment. When the sugarcane grew to the 5~6 leaf stage, the conidial suspension (with 0.05% Tween 20) (Solarbio, Beijing, China) of each experimental group was sprayed onto the young leaves of the sugarcane with a high-pressure air gun. Each experimental group had ten replicates. The control treatment was performed with sterile water. Inoculated plants were kept in a growth chamber at 28 °C with 12 h light/dark photoperiod and kept in isolation to prevent mutual infection. After 14 d of infection, the incidence rate was observed and recorded. Quantitative real-time PCR based on the *P. sorghina* var. *saccharum actin* (ACTF: TGGCAGCAGTAGTAGGAGCAGAAG, ACTR: ACGGGGACGACCGACAATGG) was performed to confirm the WT and mutant in these leaves with twisted leaf disease symptoms.

### 4.13. LC-MS/MS Method to Detect the Synthesis of Secondary Metabolites

Instruments and Reagents: The instruments and reagents included Q Exactive combined mass spectrometer (Thermo Fisher, Waltham, MA, USA); Ulti-Mate™ 3000 ultra-high performance liquid chromatography (Thermo Fisher, Waltham, MA, USA) device; ACQUITY UPLC^®^ BEH C18 column, 1.7 μm, 2.1 × 50 mm (Waters, Milford, MA, USA); MIX-200 vortex mixer (Jingxin, Shanghai, China); concentrator plus vacuum concentrator (Eppendorf, Hamburg, Germany); and 5427R centrifuge (Eppendorf, Hamburg, Germany). Methanol (chromatographically pure) and acetonitrile (chromatographically pure) were purchased from Merck Reagent Company, and formic acid (chromatographically pure) and 2-chlorophenylalanine were purchased from Thermo Fisher Company.

Sample preparation: The wild-type strain BS2-1 and the four mutant strains Δ*PsHOX1*, Δ*PsHOX3*, Δ*PsHOX6*, and Δ*PsHOX9* were cultured with Czapek Dox Liquid Medium on a shaker at 28 °C, 220 rpm for 15 d. Mycelial samples of the above strains were freshly collected. After the mycelial samples absorbed the water, we weighed 50 ± 2 mg and added 500 µL of −20 °C pre-cooled 50% methanol water (containing 1 ppm of 2-chlorophenylalanine). Subsequently, the sample was homogenized 4 times, for 30 s each time at 30 HZ. After homogenization, we shook the sample for 5 min and let it stand on ice for 15 min. We centrifuged it at 12,000 r/min for 10 min at 4 °C and pipetted 400 µL of the supernatant into a centrifuge tube. We used a 0.22 µm needle-type sterile membrane to filter the supernatant and took an appropriate amount into the lining tube of the injection bottle for LC-MS/MS analysis. We took equal amounts of each sample and mixed them as quality control samples, and tested them once before, during, and after injection. Each experimental group had three replicates.

Chromatography and mass spectrometry conditions: The conditions were as follows: ACQUITY UPLC^®^ BEH C18 column (1.7 μm, 2.1 × 50 mm), mobile phase A: 1% formic acid water, B: 100% methanol; flow rate 0.3 mL/min; gradient 0~20 min: 0~3 min (5%~20% B), 3~13 min (20%~95% B), 13~15 min (95% B), 15~18.1 min (95%~5% B), and 18.1~20 min (5% B). We used the Full-MS/dd-MS2 function for primary and secondary mass spectrometry data acquisition. The parameters were as follows: sheath gas flow: 45 L/min, auxiliary gas flow: 15 L/min; capillary temperature: 400 °C; total mass spectrum resolution: 70,000; secondary mass spectrum resolution: 17,500; collision energy: 15 eV/30 eV/45 eV (negative ion mode); spray voltage: 4.0 kV (positive ions) or −3.6 kV (negative ions).

Data analysis: We imported the raw mass spectrum data into Compound Discoverer 3.0 software for retention time correction, peak identification, peak extraction, peak integration, and peak alignment and used the database to identify and analyze the peaks containing secondary mass spectrometry data. The target database included MzCloud (Endogenous Metabolites, Natural Products/Medicines, Natural Toxins, Steroids/Vitamins/Hormones) and ChemSpider (Biocyc; ChemBank; KEGG; PlantCyc; Yeast Metabolome Database).

## 5. Conclusions

In conclusion, we sequenced the *P. sorghina* var. *saccharum* BS2-1 complete genome using Illumina and PacBio sequencing technologies. High-quality genome assembly was generated, which identified the homeobox transcription factors. We noticed that homeobox transcription factors regulate physiological processes such as growth and development, asexual reproduction, and the pathogenicity of *P. sorghina* var. *saccharum*. Among them, *PsHOX3*, *PsHOX6*, and *PsHOX9* play critical regulatory roles in the fungal growth and development, conidia formation, secondary metabolite synthesis, and pathogenicity of *P. sorghina* var. *saccharum*. The data provide an understanding of the growth and development process and pathogenic mechanism of *P. sorghina* var. *saccharum*.

## Figures and Tables

**Figure 1 ijms-25-05346-f001:**
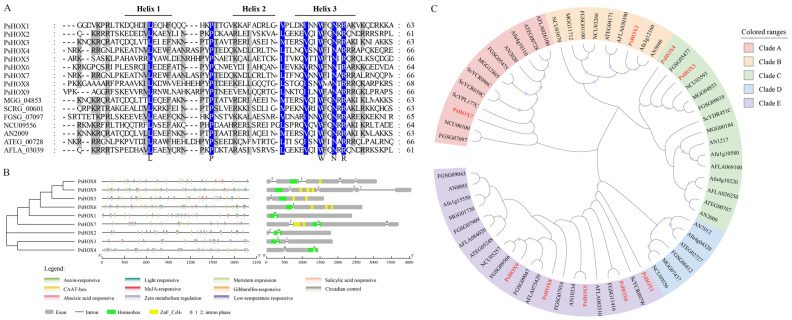
(**A**) Alignment of conserved amino acid sequence in homeobox conserved domains. Highlight homology level 100% (blue), >=75% (gray), and <75% (white). (**B**) The structure characteristics and conserved domains of homeobox transcription factor family in BS2-1. The gene structure exons, homeobox domains, ZnF-C_2_H_2_ domains, and introns are marked with gray boxes, green boxes, yellow boxes, and black lines. (**C**) Evolutionary analysis of *HOX* gene family to compare *P. sorghina* var. *saccharum*, *F. graminearum*, *A. nidulans*, *N. crassa*, *M. oryzae*, *A. flavus*, *A. fumigatus*, *A. terreus*, *A. niger*, and *A. terreus*. The red names represents the *P. sorghina* var. *saccharum HOX* gene family. The robustness of the tree was evaluated using 1000 bootstrap replicates.

**Figure 2 ijms-25-05346-f002:**
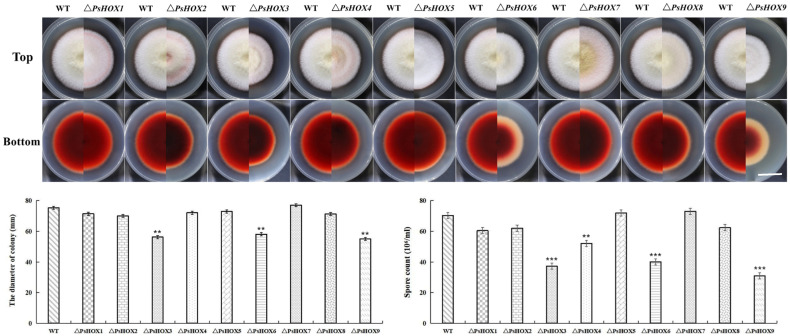
Effects of *PsHOX* genes on the wild-type strain BS2-1 (WT)’s growth and conidiophore formation. **Up**: Colonies formed by the WT and HOX mutants on PDA medium. Bars = 3 cm. **Down**: colony diameters and spore count of the indicated strains grown on PDA medium. Error bars represent the standard deviation (SD), double and triple asterisks (*) represent significant differences (*p* < 0.01 and *p* < 0.001) according to ordinary one–way ANOVA.

**Figure 3 ijms-25-05346-f003:**
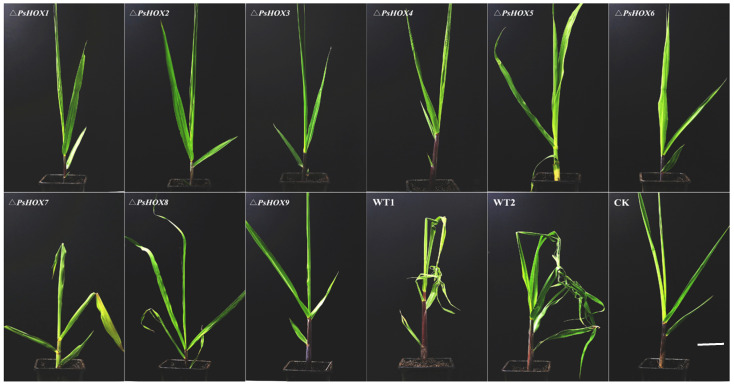
Pathogenicity test on sugarcane leaf with nine *PsHOX* gene mutants (Δ*PsHOX1*-Δ*PsHOX9*), the wild-type strain BS2-1 (positive control), and water (CK). Bars = 9 cm. Note: compare the phenotypes of sugarcane leaf after 14 d of infection.

**Figure 4 ijms-25-05346-f004:**
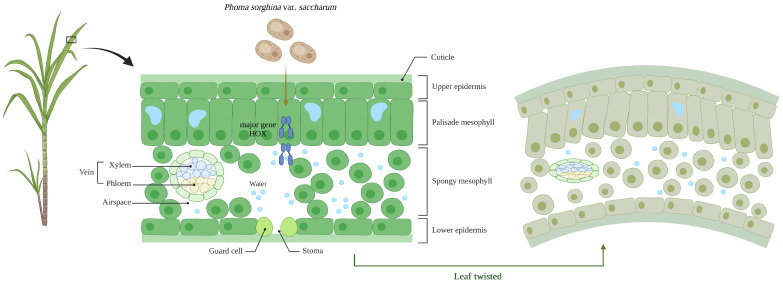
The schematic figure illustrates the intricate relations between the healthy sugarcane leaves, pathogen *P. sorghina* var. *saccharum*, and sugarcane twisted leaf disease outbreak. Box means enlarge here leaf.

**Table 1 ijms-25-05346-t001:** Characteristics of homeobox transcription factors in *P. sorghina* var. *saccharum* BS2-1 genome.

Gene Name	Gene ID	Locus	Number of Amino Acids	Molecular Weight	Theoretical PI	Instability Index	Stability	Sub-Cellular Localization
* PsHOX1 *	* PHOM_00410 *	Contig04: 763,773–766,172	783	85,593.90	6.39	61.36	unstable	Nucleus
* PsHOX2 *	* PHOM_01626 *	Contig05: 386,206–388,014	532	57,846.51	8.65	67.96	unstable	Nucleus
* PsHOX3 *	* PHOM_02470 *	Contig19: 726,424–728,280	583	63,996.62	6.38	59.55	unstable	Nucleus
* PsHOX4 *	* PHOM_03333 *	Contig00: 1,136,679–1,138,132	448	50,586.42	8.84	72.50	unstable	Nucleus or Mitochondrial
* PsHOX5 *	* PHOM_04409 *	Contig02: 1,641,196–1,642,806	521	59,044.87	6.66	52.41	unstable	Nucleus
* PsHOX6 *	* PHOM_05072 *	Contig02: 2,122,838–2,125,523	878	99,276.85	6.63	57.18	unstable	Nucleus
* PsHOX7 *	* PHOM_05197 *	Contig03: 190,707–194,407	1160	133,759.79	5.35	60.17	unstable	Nucleus
* PsHOX8 *	* PHOM_06420 *	Contig02: 1,775,319–1,778,410	882	98,336.46	5.85	53.37	unstable	Nucleus
* PsHOX9 *	* PHOM_07085 *	Contig00: 2,533,344–2,537,402	963	108,861.84	7.20	55.94	unstable	Nucleus

Note—Theoretical PI: theoretical isoelectric point; instability index: a measure of protein stability, where <40 means stable protein and >40 means unstable protein.

**Table 2 ijms-25-05346-t002:** Statistics for major secondary metabolites and categories.

Categories	Number	Secondary Metabolites
Benzene and substituted derivatives	10	4-Dodecylbenzenesulfonic acid; 3-Phenoxybenzoic acid; 3-Aminosalicylic acid; Gentian violet; 2,4-Dimethylbenzaldehyde; Benzylamine; Monobutyl phthalate; N,N’-Diphenylurea; Terephthalic acid; Dibutyl phthalate
Carboxylic acids and derivatives	15	l-Glutamic acid; l-Norleucine; l-Phenylalanine; l-Pyroglutamic acid; l-Ergothioneine; l-Histidine; N6-Acetyl-l-lysine; l-Cysteine; Citric acid; Valylproline; 4-Guanidinobutyric acid; l-Tyrosine; 4-Acetamidobutanoic acid; N-Acetyl-l-leucine; Isoleucine
Fatty Acyls	16	16-Hydroxyhexadecanoic acid; Oleic acid; Stearic Acid; Palmitic Acid; Linoleic Acid; Palmitoleic acid; Azelaic acid; Nonanoic acid; Myristic Acid; Arachidonic acid; Docosahexaenoic Acid; Dimethyl succinate; Oleamide; Acetyl-l-carnitine; Hexadecanamide; Propionylcarnitine
Organooxygen compounds	13	α,α-Trehalose; Choline; α-d-Mannose 1-phosphate; Dulcitol; d-Raffinose; d-Ribose-1-phosphate; Glucose 1-phosphate; Triethanolamine; Spermidine; Cyclohexylamine; Procyclidine; Muscone; 2-Acetylpyridine
Pyridines and derivatives	7	Picolinic acid; Nicotinamide; Nicotinic acid; 6-Hydroxynicotinic acid; Pyridoxamine; Pyridoxine; 6-Methylquinoline

## Data Availability

All data generated or analyzed during this study have been included in this published article and its Appendix A. Further inquiries can be directed to the corresponding author.

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
