# Peer review of "Genome-Wide Identification and Characterization of Homeobox Transcription Factors in Phoma sorghina var. saccharum Causing Sugarcane Twisted Leaf Disease"

_ijms, 2024, doi:10.3390/ijms25105346_

Round 1

Reviewer 1 Report (New Reviewer)

Comments and Suggestions for Authors

The authors conducted a genome-wide identification and characterization of the Hox gene family in Phoma sorghina var. saccharum, the causal agent of sugarcane twisted leaf diseases. Bao et al. identified nine Hox genes (PsHOX1–PsHOX9) in Phoma sorghina var. saccharum and investigated their knockout effects. Additionally, they observed that the growth of P. sorghina mutants was affected following the knockout of Knox genes. Furthermore, non-targeted metabolomics using LC-MS/MS on the mycelium of wild-type and four mutant strains (ΔPsHOX1, ΔPsHOX3, ΔPsHOX6, ΔPsHOX9) revealed a total of 161 secondary metabolites, categorized into 13 groups, with fatty acids being the most prevalent (16 species). Overall, while the manuscript presents valuable findings, it requires refinement.

Abstract:

The abstract could benefit from some improvements. For instance, in “line 20,” the authors mention knocking out PsHOX genes without specifying the number of knockout PsHOX genes, despite identifying nine genes. Additionally, it would be appropriate to write the first instance of “Phoma sorghina var. saccharum” in full, following the abbreviation P. sorghina var. saccharum (this should also be applied to other parts of the manuscript). Furthermore, the authors could consider including details about the methods used to knock out the Hox genes.

Line 18-19: The authors stated in “line 14” that the study identified the homeobox transcription factor family in Phoma sorghina var. saccharum at the whole genome level. Therefore, it would be better to remove “at the genome-wide level.” And avoid redundancy, which makes the redaction less concise.

Introduction

The introduction requires refinement, clarification, and reorganization. For instance, the authors mentioned in lines 33–34 that Phoma is one of the most abundant fungal genera worldwide. Later, in lines 38–39, they reiterated that "Phoma sp. is one of the most abundant... in the environment." Additionally, it is essential to double-check the references.

Line 33-44: “The genus Phoma is one of the most abundant fungal genera worldwide”. Reference?

Line 41: What do “these” refer to? Clarify

Line 42-43: which disease? Need to clarify to simplify the audience to follow.

Line 43-44: This sentence has to be rephrased

Line 53: is this the right reference?

Line 54-69: Hard to follow. The authors should polish this paragraph

Overall, the idea is there, but the introduction needs to be polished.

Results

Section 2.1: Were the genome BS2-1 assembly and BUSCO assessments conducted in this study or a previous one, “Genome Sequence of Phoma sorghina var. saccharum That Causes Sugarcane Twisted Leaf Disease in China”? Notably, the authors mentioned in lines 50-51: “We briefly reported the 50

high-quality genome of P. sorghina BS2-1”

Figure 1: Provide a high-quality image, as well for Figure 2.

Figure 3: The authors could mention in the legend how many days after knocking out the Knox genes these pictures were taken.

Material and methods

How many samples were used for each mutant to synthesize the secondary metabolites?

Comments on the Quality of English Language

can be improved

Author Response

Reviewer 2 Report (New Reviewer)

Comments and Suggestions for Authors

This study charecterized the homeobox 2 transcription factors in Phoma sorghina and showed that silencing of PsHOX genes inhibited the formation of conidia and led to a significant decrease in the pathogenicity. I suggest a major revision with the following comments.

1. Perform the comparative analysis of the BS2-1 genome of P. sorghina with previously published genomes.

2. I suggest analysing the pathogenicity of P. sorghina by qRT-PCR analysis. 

3. Add more details about LC-MS/MS non-targeted metabolomics, summarize the major metabolites, and present them in the main table.

4. Improve the discussion section by discussing your results with recent studies and adding references.

5. Add the database access date in the MM section

Comments on the Quality of English Language

Moderate editing of the English language required

Round 2

Reviewer 2 Report (New Reviewer)

Comments and Suggestions for Authors

The authors improved the manuscript significantly, and it looks better than before. I recommend the manuscript for publication to IJMS.  

This manuscript is a resubmission of an earlier submission. The following is a list of the peer review reports and author responses from that submission.

Round 1

Reviewer 1 Report

Comments and Suggestions for Authors

The manuscript by Bao et al reports the genome sequence of Phoma sorhina BS-21, which is a pathogen of sugarcane, Gene annotation, Identification of gene clusters for secondary metabolite production, deletion of HOX family genes, and phenotype analysis of these mutants. This study provided solid data on P. sorghina BS2-1. However, this significance needed to be better demonstrated.

1. The abstract lacks an essential description of Phoma sorghina var sacchaum and why it is so important. The relationship between Phoma sorghina var sacchau and P. sorghina BS2-1 is also unclear. P. sorghina BS2-1 must be submitted to some biobanks for future verification.

2. Section 2.1. There is no explanation for an improvement from author’s previous study [8], which already reported a high-quality genome of P. sorghina BS2-1. The list of contigs and number of genes shown in Tables 1 and 2 should be moved to the supplementary because these are not the main topic of this study.

3. Section 2.3 has very poor link with other parts of the manuscript. The metabolomic analysis confirming the prediction for the capability of secondary metabolism production is at least essential.

4. Sections 2.4 and 5 lack an explanation of why HOX genes are so crucial for pest control purposes.

5. Section 2.6 and Fig. 3 also should be moved to the supplementary because it only explains the method to construct the gene knock-out mutants and its confirmation by the PCR analysis

6. The section 2.7 looks interesting, but additional phenotype data, such as the metabolic profile data of secondary metabolites, is needed to investigate the relationship between the HOX genes and pathogenicity

7. Section 2.8 questioned whether the loss of pathogenicity might be derived from some unknown effect derived from the gene manipulation procedure because all mutant strains showed very similar phenotypes. A negative control (a mutant strain lacking another useless gene), which shows the pathogenicity, is needed.

8. The discussion lacks a justification for this study. It is still being determined how the findings demonstrated in this study will be helpful for developing pest control methods to protect sugar cane from P. sorghina BS2-1.

Author Response

  • The abstract lacks an essential description of Phoma sorghinavar. sacchaum and why it is so important. The relationship between Phoma sorghina var. sacchaum and sorghina BS2-1 is also unclear. P. sorghina BS2-1 must be submitted to some biobanks for future verification.

A: Thank you for your comments. We added the essential description of Phoma sp. and its importance. Furthermore, we explained the relationship between Phoma sorghina var. sacchaum and P. sorghina, and also provided the Genbank database accession number.

  • Section 2.1. There is no explanation for an improvement from author’s previous study[8], which already reported a high-quality genome of sorghinaBS2-1. The list of contigs and number of genes shown in Tables 1 and 2 should be moved to the supplementary because these are not the main topic of this study.

A: Thank you for your suggestions. Table 1 and Table 2 have been moved to the supplementary according to the suggestion, and we explained the high-quality genome significance.

  • Section 2.3 has very poor link with other parts of the manuscript. The metabolomic analysis confirming the prediction for the capability of secondary metabolism production is at least essential.

A: Thank you for your comments. The identification and characterization of secondary metabolite genes (SMs), is part of the comparative genomic analysis of Phoma sp. One of the objectives of the current study were in-depth molecular and genomic analysis of twisted leaf diseases, and to reveal the differences between the genome of the sequenced strain of P. sorghina BS2-1 with three reference genomes of SMs. According to the suggestions, we will further perform metabolomic analyses to reveal the ability of SMs production by P. sorghina.

  • Sections 2.4 and 5 lack an explanation of why HOX genes are so crucial for pest control purposes.

A: Thank you for your comments. Previous work has shown that the HOX family gene of fungi is mainly involved in regulating mycelial polar growth, sexual reproduction, sporulation, and pathogenicity. Deleting the homeobox gene GRF10 in Candida albicans inhibited mycelial growth, spore defects, and biofilm formation. Moreover, the HD-Zip might regulate the leaf curled and twisted. According to the reviewer’s suggestions, we have added the detailed account of HOX genes in the introduction section.  

  • Section 2.6 and Fig. 3 also should be moved to the supplementary because it only explains the method to construct the gene knock-out mutants and its confirmation by the PCR analysis.

A: Thank you for your suggestions. The Fig. 3 has been added in supplementary data according to the suggestions.

  • The section 2.7 looks interesting, but additional phenotype data, such as the metabolic profile data of secondary metabolites, is needed to investigate the relationship between the HOX genes and pathogenicity.

A: Thank you for your valuable remarks. Earlier studies have proved that HOX genes play an essential role in fungal asexual reproduction. In C. albicans, the HOX gene GRF10 regulates mycelial growth. Knockout of HOX gene (pah1) in P. anserina slowed hyphal growth and abnormal colony morphology. After the deletion of HOX2 in F. graminearum, the development, and synthesis of conidia were inhibited. HOX2 plays a crucial role in the spore development of M. oryzae, and the loss of MoHOX2 causes M. grisea to lose the ability to form conidia. In this study, our experimental results demonstrated that HOX family genes play a crucial role in regulating the formation of mycelium and conidia, as well as the pathogenicity of P. sorghina. Further, we will perform the metabolomic analysis to reveal the in-depth relationship between the production of SMs and pathogenicity in P. sorghina.

  • Section 2.8 questioned whether the loss of pathogenicity might be derived from some unknown effect derived from the gene manipulation procedure because all mutant strains showed very similar phenotypes. A negative control (a mutant strain lacking another useless gene), which shows the pathogenicity, is needed.

A: Thank you for your valuable remarks. Sugarcane twisted leaf disease is a fungal disease caused by P. sorghina. Symptoms include twisting and curling of the leaves, converting all twisted leaves into a mass in later stages.In this study, pathogenicity test on sugarcane leaf with nine HOX gene mutants (ΔPsHOX1-ΔPsHOX9), including the wild-type strain BS2-1 (positive control) and water (negative control). The sugarcane leaves inoculated with nine mutants showed that there are various degrees of mild leaf disease symptoms and the loss of twisted symptoms, compared with the control. It shows that the knockout of HOX genes led to the different level of reduction in the pathogenicity of the mutants.

  • The discussion lacks a justification for this study. It is still being determined how the findings demonstrated in this study will be helpful for developing pest control methods to protect sugar cane from sorghinaBS2-1.

A: Thank you for your comments. The discussion has been revised according to the suggestions. Currently, practical significance for sugarcane industry is to strengthen the applied basic research on the prevention and control of sugarcane leaf curl disease. It will lead to excavation of key disease-causing factors of the pathogenic bacteria and will provide the molecular basis for the subsequent elimation of resistance genes in sugarcane. Furthermore, the selection and breeding of resistant varieties will pave a path for the fundamental solution to the sugarcane tip-rot disease prevention and control.

Reviewer 2 Report

Comments and Suggestions for Authors

The MS reflects research carried out with a foundation requiring a lot of research work and serious financial investment. The set tasks have been implemented. The research planning, implementation and discussion of the results are correct, with many and demanding illustrations.

Minor errors:

ad 16 the majority of eukaryotes do not have embryonic development (suggestion: ….among others…)

ad 32-33 abiotic and biotic diseases ?

ad fig 1. According to nomenclature regulations, prefixes (here: sp.) do not have to be highlighted.

gives Table 3. The table deserves more explanation. For example: what is PI or instability index?

ad 258 & 259: Are the references correct?

ad 265 Where are the data on reducing the number of conidia?

ad fig 6: its explanation requires a language check

Ad 464: Please write Phoma correctly.

ad 489 Correctly written: in Collectotrichum Scoville-Pepper …?

It might be worth explaining the following abbreviations: CDS, gDNS, HOX

Comments on the Quality of English Language

High quality MS with smaller problems.

Author Response

1) Q: ad 16 the majority of eukaryotes do not have embryonic development (suggestion: ….among others…)

A: Thank you for highlighting the ambiguities in the manuscript. We have carefully revised the manuscript, according to your comments and suggestions.

  • Q: 32-33 abiotic and biotic diseases ?

A: Thank you for the suggestions. We have changed the statement to abiotic and biotic factors.

  • Q: ad fig 1. According to nomenclature regulations, prefixes (here: sp.) do not have to be highlighted.

A: Thank you for your valuable remarks. Fig 1 have been revised according to the proposal.

  • Q: gives Table 3. The table deserves more explanation. For example: what is PI or instability index?

A: Thank you for your valuable remarks. Table 3 have been revised according to the proposal. We increase the explanation, including theoretical PI and instability index.

  • Q: ad 258 & 259: Are the references correct?

A: Thank you for highlighting the ambiguities in the manuscript. We have carefully checked the manuscript references and revised.

  • Q: ad 265 Where are the data on reducing the number of conidia?

A: Thank you for your valuable remarks. The Fig 4 histogram has been shown that the number of conidia is decrease.

  • Q: ad fig 6: its explanation requires a language check

A: Thank you for your valuable remarks. The Fig 6 explanation has been revised according to the suggestions.

  • Q: ad 464: Please write Phoma correctly.

A: Thank you for highlighting the ambiguities in the manuscript. We have carefully checked the manuscript spelling and revised them.

  • Q: ad 489 Correctly written: in Collectotrichum Scoville-Pepper …?

A: Thank you for highlighting the ambiguities in the manuscript. We have carefully checked the manuscript typeface and revised them.

10) Q: It might be worth explaining the following abbreviations: CDS, gDNA, HOX

A: Thank you for your valuable remarks. The abbreviations (e.g. CDS, gDNA, HOX) has been revised according to the suggestions.